



# Automated compound speciation, cluster analysis, and quantification of organic vapours and aerosols using comprehensive two-dimensional gas chromatography and mass spectrometry

Xiao He[1], Xuan Zheng[1*], Shuwen Guo[1], Lewei Zeng[1], Ting Chen[1], Bohan Yang[1], Shupei Xiao[1], Qiongqiong Wang[2], Zhiyuan Li[3], Yan You[4], Shaojun Zhang[5,6,7,8], and Ye Wu[5,6,7,8]

[1]College of Chemistry and Environmental Engineering, Shenzhen University, Shenzhen 518060, China

[2]Department of Atmospheric Science, School of Environmental Studies, China University of Geosciences, Wuhan 430074, China

[3]School of Public Health (Shenzhen), Sun Yat-sen University, Guangzhou 510275, China

[4]National Observation and Research Station of Coastal Ecological Environments in Macao, Macao Environmental Research Institute, Macau University of Science and Technology, Macao SAR 999078, China

[5]School of Environment, State Key Joint Laboratory of Environment Simulation and Pollution Control, Tsinghua University, Beijing 100084, China

[6]State Environmental Protection Key Laboratory of Sources and Control of Air Pollution Complex, Beijing 100084, China

[7]Beijing Laboratory of Environmental Frontier Technologies, School of Environment, Tsinghua University, Beijing 100084, China

[8]Laboratory of Transport Pollution Control and Monitoring Technology, Transport Planning and Research Institute, Ministry of Transport, Beijing 100028, China

*Correspondence to*: Xuan Zheng (x-zheng11@szu.edu.cn)



**Abstract:** The advancement of analytical techniques, such as comprehensive two-dimensional gas
chromatography coupled with mass spectrometry (GC×GC-MS), enables the efficient separation of
complex organic matrix. Developing innovative methods for data processing and analysis is crucial to
unlock the full potential of GC×GC-MS in understanding intricate chemical mixtures. In this study, we
proposed an innovative method for the semi-automated identification and quantification of complex
organic mixtures using GC×GC-MS. The method was formulated based on self-constructed mass
spectrum patterns and the traversal algorithms and was applied to organic vapor and aerosol samples
collected from tailpipe emissions of heavy-duty diesel vehicles and the ambient atmosphere. Thousands
of compounds were filtered, speciated, and clustered into 26 categories, including aliphatic and cyclic
hydrocarbons, aromatic hydrocarbons, aliphatic oxygenated species, phenols and alkyl-phenols, and
heteroatom containing species. The identified species accounted for over 80% of all the eluted
chromatographic peaks at the molecular level. A comprehensive analysis of quantification uncertainty
was undertaken. Using representative compounds, quantification uncertainties were found to be less than
37.67%, 22.54%, and 12.74% for alkanes, polycyclic aromatic hydrocarbons (PAHs), and alkyl-
substituted benzenes, respectively, across the GC×GC space, excluding the first and the last time
intervals. From source apportionment perspective, adamantane was clearly isolated as a potential tracer
for heavy-duty diesel vehicles (HDDVs) emission. The systematic distribution of N-containing
compounds in oxidized and reduced valences was discussed and many of them served as critical tracers
for secondary nitrate formation processes. The results highlighted the benefits of developing self-
constructed model for the enhanced peak identification, automated cluster analysis, robust uncertainty
estimation, and source apportionment and achieving the full potential of GC×GC-MS in atmospheric
chemistry.



**1 Introduction**

Improved sampling strategies, coupled with innovative measurement techniques, are imperative to capture the dynamic nature of atmospheric chemistry, particularly in the context of climate change and health risk (Franklin et al. 2023, Franklin et al. 2022, Huo et al. 2021, Phillips et al. 2018). Comprehensive two-dimensional gas chromatography coupled with mass spectrometry (GC×GC-MS) has emerged as a powerful tool for compound detection and identification, benefitting from the combination of two columns with orthogonal selectivity (Alam et al. 2013, Franklin et al. 2022).

Despite its capabilities, GC×GC-MS encounters formidable challenges in data analysis, which can be extremely complicated and demanding. Efforts have been made to delve into the deluge of data generated by GC×GC-MS. Traditionally, mass spectra were deconvoluted and compared to spectra from the National Institute of Standards and Technology (NIST20) library for peak identification with pre-defined criteria (Guo et al. 2016, Piotrowski et al. 2018). Retention indices (RI) were further introduced to distinguish homologous compounds with resembling mass spectra (Zang et al. 2023). A pioneering and instructive work for searching criteria to classify GC×GC peaks was published in 2003 (Welthagen, Schnelle-Kreis and Zimmermann 2003). Welthagen (2003) incorporated the mass fragmentation patterns to classify compounds in atmospheric aerosol samples. Compounds belonging to the same chemical group related to one another in the GC×GC space and distributed in a structured pattern. They successfully identified seven groups of compounds, including alkanes, alkenes and cycloalkanes, alkyl substituted benzenes, alkyl substituted polar benzenes, hydrated naphthalenes and alkenyl benzenes, alkylated naphthalenes, and alkane acids, occupying more than 60% of the total peak area. This work set a good example of how user-defined rule could facilitate the identification of specific compound groups. Recent advances in chemometric tools for GC×GC-MS analysis involving machine learning and deep learning renovate multi-dimensional chromatography fields (Stefanuto, Smolinska and Focant 2021). Bendik (2021) developed a programming suite for high-confidence and fast compound identification using GC×GC coupled with time-of-flight mass spectrometry (TOF-MS) (Bendik et al. 2021). He (2022) extracted featured mass spectrometric information of the intermediate-volatility and semi-volatile organic compounds (I/SVOCs) by integrating algorithmic language into GC×GC-MS data (He et al. 2022a, He et al. 2022b). A novel pixel-based multiway principal component analysis method was utilized in Song (2023) to identify key tracers during incense burning (Song et al. 2023). Nevertheless, the



interpretation of GC×GC-MS data demands advanced computational tools and expertise, and the
investigation of unknown compounds remains scarce due to the inadequate validation procedures,
overreliance on manual data processing, limited access to computational capabilities, and the lacked
expertise to handle the complex chromatographic data effectively.
Bridging this gap requires further development of sophisticated algorithms and analytical approaches to
unlock the full potential of GC×GC. This study proposed a bottom-up method for cluster analysis and
quantification of organic vapours and aerosols within complex atmospheric mixtures. The scripts were
initiated with the recognition of the common mass spectra features of specific species and tailored to a
wide range of compound clusters. The scripts were then trained, iterated, and optimized incorporating
real sample data until robust outputs were achieved. The new strategy reduced the ambiguity that is often
associated with identifying compounds in complex mixtures.
The proliferation of heavy-duty diesel vehicles (HDDVs) has raised significant concerns, with an
escalating demand for freight transport and in various industrial operations (Yan et al. 2022, Cheng et al.
2022). Despite a low retention rate, HDDVs release massive amounts of particulate matter, nitrogen
oxides, ammonia, and carbon monoxide into the atmosphere, compared with other vehicle types (Wang
et al. 2023, Silva et al. 2023, Chang et al. 2022, Stanimirova et al. 2023, Hamilton and Harley 2021,
2021, Kruve et al. 2014). Given this, the gas and aerosol samples from representative HDDV tailpipes
and the ambient environment were collected and analyzed by GC×GC-MS. The proposed bottom-up
method was applied for a comprehensive analysis of the complex organic mixtures, resolving 26
compound categories including hydrocarbons in multiple forms, oxygenated components, and
heteroatom containing species. Over 80% of all the chromatographic peaks were identified and assigned
to a compound cluster using proposed method, leaving a minor portion of organic matrix unresolved.
Different compound clusters occupied separate positions in the GC×GC space, and distinctive
distribution patterns within diverse samples and their contribution fractions were revealed. Quantification
uncertainties were addressed thoroughly and the significant potential deviation when using n-alkanes as
semi-quantification surrogates was proved. Overall, the integration of automated algorithms and GC×GC
data analysis holds significant implications for advancing our understanding of atmospheric chemistry,
improving secondary organic aerosol (SOA) estimation, and thus guiding the implementation of
environmental policies.



## 2 Materials and methods

### 2.1 Sample collection, treatment, and instrumental analysis

For the collection of HDDVs tailpipe emission, chassis dynamometer experiments were conducted at the China Automotive Technology & Research Center (CATARC) in Guangzhou, China. Exhaust emissions from HDDVs were diluted in a constant volume sampler (CVS, CVS-ONE-MV-HE, Horiba), following the China heavy-duty commercial vehicle test cycle for tractor trailers (CHTC-TT) driving cycles. The average temperature in the sampling train was precisely controlled at 47 °C, and airflow, relative humidity, and airflow, relative humidity, and pressure were monitored simultaneously. The speed trace and characteristics of CHTC-TT are shown in Figure S1.

Gaseous exhausts were collected by two adsorbent thermal desorption (TD) tubes in series (Tenax TA, C1-AXXX-5003, Markes International) after being filtered by a Teflon filter. Particulate exhausts were deposited on a 47 mm quartz filter (Grade QM-A, Whatman). Ambient $PM_{2.5}$ filter samples were collected on the rooftop of a 5-story building on the campus of Shenzhen University (22.60°N, 114.00°E) in western Shenzhen, approximately 25 m above the ground. The sampling site was surrounded by campus, residential areas, greenbelts, and a golf park and, the location map is shown in Figure S2. Previous studies demonstrated that the $PM_{2.5}$ concentration in this aera represented the average pollution scheme in Shenzhen (Huang et al. 2018, Yu et al. 2020). Sampling strategy followed a regular schedule of one 24-h sample every day using a high-volume sampler (Th-1000c II, Wuhan Tianhong Environmental Protection Industry Co., Ltd). TD samples were kept dry at room temperature, and quartz filters were stored frozen at −18 °C before analysis. All sampling materials were pre-baked thoroughly to remove potential carbonaceous contamination.

TD samples were injected with 2 µL of deuterated internal standard (IS) mixing solution through a mild $N_2$ blow (CSLR, Markes International). A precious portion of 1 cm$^2$ (1 cm × 1 cm) filter sample was isolated and cut into strips. They were spiked with 2 µL of IS mixing solution and inserted into a passivated quartz tube. All TD samples and quartz tubes were loaded onto a thermal desorption autosampler (ULTRA-xr, Markes International), thermally desorbed (UNITY-xr, Markes International), and subjected to GC×GC separation (Agilent 8890, Agilent Technologies; Solid State Modulator1810, *J&X* Technologies) and mass spectrometry detection (Agilent 5977B, Agilent Technologies).



132 The thermal desorption system heated the TD tubes to 320 °C (quartz tubes to 330 °C) for 20 min, while

133 the trap remained at 20 °C. Following tube desorption, the trap temperature was raised to 330 °C (340

134 °C for quartz tubes measurement) for 5 min at the maximum heating rate, and the vaporized analytes

135 were purged into the 1st GC column with a desorb split flow of 6 mL/min. Separation of the analytes was

136 carried out using a DB-5ms capillary column (30 m × 0.25 mm × 0.25 μm, Agilent Technologies) as the

137 primary column and a DB-17MS capillary column (1.2 m × 0.18 mm × 0.18 μm, Agilent Technologies)

138 as the secondary column. The modulation column consisted of a VF-1MS capillary column (0.7 m × 0.25

139 mm × 0.10 μm, Agilent Technologies) connecting to the 1st column and an Ultimate Plus deactivated

140 fused silica tubing (0.6 m × 0.25 mm, Agilent Technologies) connecting to the 2nd column.

141 Initially, the GC oven was set at 50 °C for a 3-min duration, followed by a gradual increase at a rate of 5

142 °C/min until it reached 310 °C, where it was maintained for an additional 5 minutes. The entry and exit

143 hot zones were +10 °C higher than the GC oven temperature, and the trap zone was maintained at -50

144 °C. The modulation cycle had a period of 4 s. Carrier gas flow was set at 1.2 mL/min. The ion source

145 was kept at 250 °C and scanned over a range of 20 – 350 amu.

146 **2.2 Data collection, alignment, and parsing**

147 GC×GC-MS data acquisition was conducted using Enhanced MassHunter (version 10.0, Agilent

148 Technologies) and SSCenter (version 2.4.0.0, *J&X* Technologies). All data utilized to develop and test

149 the scripts were processed by Canvas Browser (version 2.5, *J&X* Technologies) for basic preprocessing,

150 such as baseline correction, mass spectra deconvolution, and peak smoothing. The application of baseline

151 correction and peak smoothing allowed for an increased signal-to-noise ratio (S/N) and improved overall

152 data quality. Chromatographic peaks were filtered according to the filtering rules: baseline noise = 150,

153 S/N > 50. For each individual sample, after isolating all compounds of interest, a peak table in 1st

154 retention time (RT) sequential order with 1st RT and 2nd RT, peak area, peak height, peak width, and

155 deconvoluted mass spectra was exported. These quantitative variables were further processed for targeted

156 and non-targeted "omics" oriented analysis.

157 As anticipated, the chromatographic variables experienced RT shifts due to column degradation, routine

158 maintenance (e.g., cutting column), and system fluctuation (e.g., carrier gas pressure variation). The

159 initial RT shifting tolerance for adaptive cluster matching was set to be 1 period of modulation in the 1st

160 dimension and 0.1 s in the 2nd dimension. Additionally, a 2D shift cluster consisting of $C_{16}D_{34}$, $C_{24}D_{50}$,



and $C_{32}D_{66}$, with the merit of correcting $2^{nd}$ RT shift, was configured. Data correction or data alignment
is crucial for accurate and consistent peak integration.

**2.3 Algorithmic development**

EI spectra are typically characterized by a molecular ion ($M^+$) peak plus a collection of fragment ion
peaks. The $M^+$ may dominate the mass spectrum in some cases (e.g., un-substituted polycyclic aromatic
hydrocarbons (PAHs)), and more frequently presents at a relatively low intensity. The EI spectra are
highly comparable among different instrument systems and experimental conditions, making them an
excellent measure to identify compounds. The characteristic ions and their relative intensities depend on
the intrinsic nature of the targeted compounds, necessitating knowledge of basic rules and common
fragmentation routes to interpret EI mass spectra. Figure 1 illustrates the workflow for establishing
computational strategies for robust and reproducible GC×GC-MS data processing.
Functional groups have significant effects the fragmentation patterns observed in mass spectrometry, and
some ions are typical of given structures. Isotopic peaks (e.g., hydrogen and chlorine) provide additional
information about the molecules (Du and Angeletti 2006, Fernandez-de-Cossio et al. 2004). These pieces
of information formed the foundation for building up the model for cluster analysis and are addressed in
greater detail in the supporting information (S1). These indicative reaction schemes have been
incorporated into the model development. Each critical step of model construction and validation is
described thoroughly. The quantitative variables in the data alignment table, combining the
chromatographic and MS information are properly exploited and determine the overall speciation
capacities. Traditionally, compound identification relies on the electron ionization-based fragmentogram
and the deconvoluted mass spectra. Empirically speaking, one-by-one compound identification would
be greatly intervened by neighbouring peaks, especially those with similar structures, and introduce
considerable uncertainties. A good example is the assignment of homologous *n*-alkanes, of which the
fragmentogram bears a close resemblance (Figure S5). In such cases, the similarity score (the measure
of similarity between the observed mass spectrum and the NIST library hit) could be erroneously inflated
to 850 (out of 999) or higher. In contrast, cluster analysis or "omics" oriented analysis involves the
comprehensive analysis of a specific type of compounds on a large scale, aiming to provide a holistic
understanding of the distribution and transformation of the specific compound cluster being investigated.



Due to the complexity and remarkable peak capacities, sophisticated and detailed scripts for cluster
identification were constructed. Heteroatom containing species, e.g., amides and amines, were carefully
examined. The scripts began by recognition of the common mass spectra features of compound cluster
of interest and are addressed in more details as follows:

193       1.   The Boolean value of characteristic ions.

194       2.   The intensity sequence of abundant ions in the whole spectra.

195       3.   The retention time window restriction for certain compound groups.

196       4.   The pattern of mass spectrometry variation with the increased number of substituents or the

197            extension of the carbon chain.

198       5.   Iteration framework that involved repetitive cycles among all the tested samples.

The scripts were then trained, iterated, and optimized incorporating real sample data, and the parameters
were adjusted accordingly until a robust output was achieved. The extractor function built in the Canvas
software was activated, and all the scripts were imported to facilitate automated cluster analysis. The
scripts parsed all the files in the given directory into the required structure and generated three reports in
the form of .pdf, .csv, and .bmp. The .csv file contained key information including the compound name,
compound cluster, 1st and 2nd RTs, and peak area (based on total ion current (TIC)).
Once exported, the peaks were further processed for quantification/semi-quantification following the
steps below. First, calibration curves were prepared by spiking different volumes of the standard solution
mixture onto the blank TD tubes and blank filters, respectively. Peak area ratios, i.e., peak area of
authentic standards over that of the internal standards, were used to build the linear relationship, with the
merit of correcting system fluctuation. The selection of authentic standards prioritizes their wide
distribution across the entire chromatogram space, ranging from high to low volatility and weak to strong
polarity, and meanwhile encompassing a broader range of functional groups and heteroatoms. The
distribution and performance of all authentic standards are summarized in Table S1 and Figure S6.
Second, for the un-quantified peaks, their complied information (X, Y, Z) corresponding to (1st RT, 2nd
RT, compound cluster) is looped through the list of all authentic standards in the following descriptive
algorithm framework (note that the statements do not conform to the grammar rule and it is for illustrative
purpose only) until the optimal authentic standard to semi-quantity the target peaks is exported. It should
be emphasized that the un-quantified peak and the corresponding authentic standards to semi-quantify it
must belong to the same group, due to their physics-chemical similarities.




For (i = 1 to n) # n equals the number of authentic standards and is a known variable.

If (ZM = Zi) # M is the un-quantified peak and i refers to the authentic standard that is selected in a certain loop.

Ai = Min (an array of ((XM – Xi)$^2$ + (YM – Yi)$^2$)) # This sentence dose not conform to the grammar rule of Visual Basic for Applications in Excel, and it is for illustrative purpose only.

Export Zi, (Xi, Yi, Zi), its peak area, and its linear calibration relationship.

End if

Next i


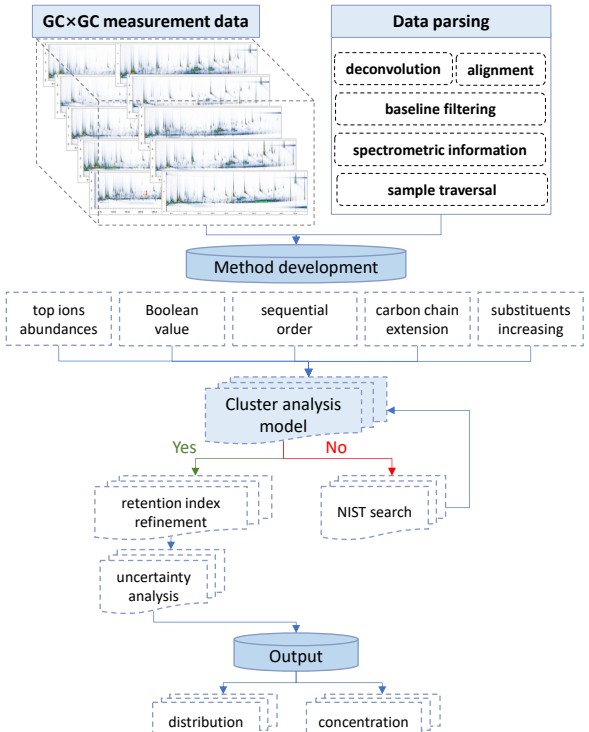

**Figure 1. Flow diagram illustrating the multistep data processing for establishing computational strategies**
**for cluster analysis and quantification of organic vapours and aerosols using GC×GC-MS data.**
**2.4 Quality assurance/control and uncertainty evaluation**
It is common for thermal decomposition to occur in analytical methods involving heating processes,
potentially leading to the erroneous detection of compounds that are either not present in real samples or



present in low concentrations. Such artifacts need careful scrutiny, and the availability of authentic
standards covering the GC×GC space range is essential for validation. The thermal programs used in this
study were highly similar to those employed by Franklin (2023), who observed low chromatographic
signals of decomposition products (Franklin et al. 2023). Nevertheless, the possibility that some observed
analytes are decomposition products cannot be entirely ruled out. Peaks of ISs were traced across all
samples to monitor the variation of several modules, and the results are presented in Figure S7. Excellent
stability was clearly observed, demonstrating the robustness of the testing system. Strong linear
correlations were achieved for this set of authentic standards, with Pearson's R ranging from 0.97 to
0.99. Routine blank tests were conducted to prevent unexpected contamination.
**3 Results and discussion**
**3.1 Overall performance of the algorithm and compound identification**
The optimization of component identification remains a challenging issue, and this work involves
converting known chemical compounds into molecular descriptors and utilizing cluster analysis to
predict the relationship between these descriptors and structural information. After continuous trails to
improve reliability and data processing speed, a final solution of 26 compound clusters stands out with
high accuracy and repeatability:
– Aliphatic hydrocarbons, including *n-/i-* alkanes and alkenes
– Cycloalkanes
– Alkyl-substituted benzenes, including $C_1 – C_6$ alkyl-substituted benzenes
– Admantanes
– Hopanes
– 2 – 5 ring PAHs
– Acids
– Aliphatic alcohols
– Aliphatic aldehydes and ketones
– Oxy-PAHs
– Phthalates
– Phenols and alkyl-substituted phenols



–    Phenol ethers
–    Amides
–    Amines
–    Pyridines
–    Nitros, including organic nitrates and organic nitrites
Validation of the model output using field diesel samples has been conducted and has shown high
estimation accuracy and integrity. Generally, over 82% of the peaks have been successfully classified
and assigned to the corresponding compound groups and their distribution in an example GC×GC plot
is shown in Figure 2. To confirm the tentatively identified heteroatom groups, their raw chromatogram,
mass spectra, and chemical structures of representative species are displayed in Figures S9-S15. Less
than 18% of the chromatographic peaks were identified as unresolved components. Basically, aliphatic
hydrocarbons were located in the lowest position in a GC×GC space except column bleedings (Figure
2a-c and Figure S8), and their $2^{nd}$ RT drift was less than 1s from the far-left to the far-right side. N-
containing compounds in oxidized and reduced valences, including amides, amines, pyridines, and nitros,
were resolved simultaneously under respective filtering rules, and they occupied a slightly higher
position in the GC×GC space (Figure 2f). Amines and pyridines were more volatile species and eluted
at early stages, whereas nitros and amides were eluted at middle and late stages sequentially. Due to their
high volatility, $C_2$-$C_6$ alkyl-substituted benzenes also presented at the beginning of the GC×GC space
and they partitioned dominantly into the gas phase. Their $2^{nd}$ RTs were comparable to those of pyridines
and amides, and the $2^{nd}$ RT drift was negligible. Aliphatic O-containing compounds, including acids,
alcohols, and ketones, were found to be in the middle region and covered a wide volatility rang. Aliphatic
O-containing compounds affect the acidity of the atmosphere, participate in aqueous phase reactions, and
contribute significantly to the formation of SOA (Cope et al. 2021, Xu et al. 2022). Phenols with one or
more hydroxyl groups attached to an aromatic benzene ring were observed in the middle of the GC×GC
space. Oxy-PAHs and PAHs presented in the upper middle of GC×GC space, and the volatility range
stretched to the low volatility end, for which a clear trend tilting to the upper right corner was observed,
suggesting that the aromaticity played a significant role in the retention in the secondary dimension.



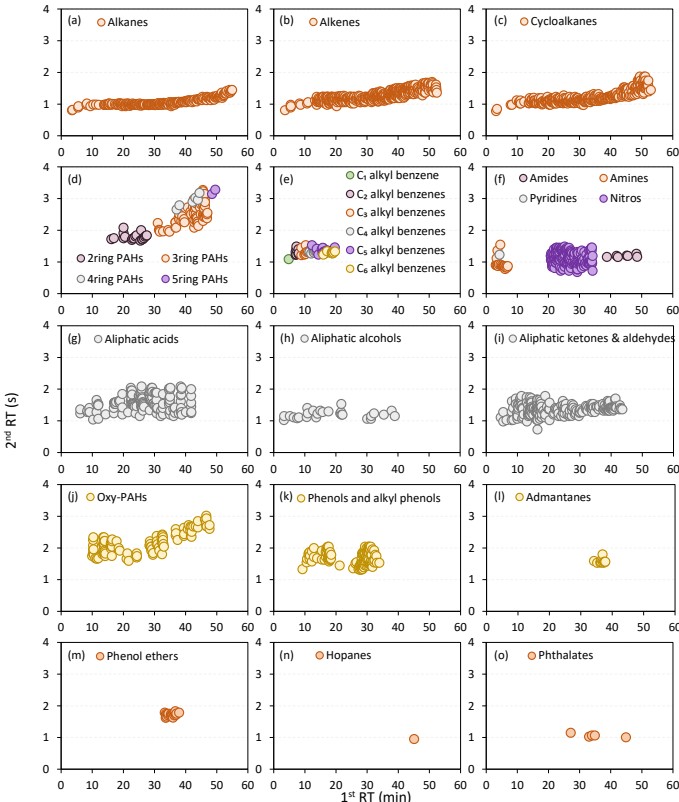

**Figure 2. The distribution of the 26 compound groups in an example GC×GC plot. For clear visualization, different compound groups are displayed separately, except for 2 – 5 ring PAHs, $C_2$ – $C_6$ alkyl-substituted benzenes, and N-containing species. Nitros include organic nitrates and organic nitrites, due to the co-existence of the characteristic ions at m/z 30 ($NO^+$) and m/z 46 ($NO_2^+$).**

**3.2 Model uncertainty estimation**

We conducted a systematic evaluation of the model output, and the results are shown in Figure 3 and Figure 4. To address this issue comprehensively and accurately, we selected three types of standards including $C_7$ – $C_{37}$ *n*-alkanes, $C_2$ – $C_6$ alkyl-substituted benzenes, and 2 – 4 ring PAHs, representing a full range of low to high polarity and various functionalities. The quantification deviation was computed according to the principles of the model. Chromatographic peaks were quantified by either their authentic standards or the surrogates that fell within the same compound category after being classified to one of the 26 compound classes. For example, if the mass spectrum of a chromatographic peak followed the pattern of the compound class of alkanes, it would be assigned into the alkane group and quantified by its authentic standard if any, or by the *n*-alkane (*n*-alkane serving as the semi-quantification surrogate in



this case) that was closest to it spatially. Similarly, if the mass spectrum of a chromatographic peak
followed the pattern of $C_x$ alkyl-substituted benzenes, it would be assigned into the $C_x$ alkyl-substituted
benzene group and quantified by its authentic standard if any, or by the alkyl-substituted benzene (alkyl-
substituted benzenes serving as the semi-quantification surrogate) that was closest to it spatially. In light
of the explanation, the deviation of the slopes of the calibration curves of any pair of the adjacent
authentic standards that fell within the same compound category was computed to represent the ceiling
of the semi-quantification uncertainty. Uncertainties are computed using the following Eq. (1):
$$Uncertainty\ (\%) = \frac{Abs(S_p - S_s)}{Smaller\ (S_p, S_s)} * 100 \tag{1}$$
where $S_p$ and $S_s$ are the slopes of the previous and subsequent compounds, respectively.
The slopes increased rapidly from 3.13 ($C_7$ $n$-alkane) to 8.21 ($C_9$ $n$-alkane), fluctuated slightly from 8.85
to 11.8 in the range of $C_9$ to $C_{27}$ $n$-alkanes, and decreased gradually after $C_{28}$ $n$-alkane to the end of $C_{37}$
$n$-alkane. In the whole volatility range of $C_9 - C_{37}$ $n$-alkanes, the uncertainties were less than 37.67%,
except for one time interval between $C_8$ and $C_9$ $n$-alkanes, where the quantification deviation reached
142%. A similar trend was observed for PAHs, with uncertainties less than 22.54%, except for the first-
and last-time intervals, where the quantification deviations were 55.44% and 81.59%, as shown in Figure
3. Stable responses of $C_2 - C_6$ alkyl-substituted benzenes were monitored, and the uncertainties were less
than 12.74%. In other words, for any given peak, it would be quantified/semi-quantified by one authentic
standard, and the upper limit of quantification uncertainty originated from any pair of the adjacent
authentic standards was as discussed earlier.
It made sense that the uncertainty ranges of alkyl-substituted benzenes were less than those of $n$-alkanes
and PAHs, given that alkyl-substituted benzenes were eluted early at the front half, whereas alkanes and
PAHs covered the whole volatility range. The trends illustrated that the responses of GC×GC to the
analysts were sensitive to the volatility distribution and region for accurate quantification fell in the
middle part. It also highlighted the utility of introducing more authentic standards and the benefits of
enriching compound categories. We could speculate that the quantification uncertainty would be further
reduced with the augmentation of standard compounds.
Furthermore, we delved into the uncertainty estimation of dividing the whole chromatogram into bins
based on retention time, and all the species in the same bin were quantified, referring to the mass-to-
signal responses of the $C_n$ $n$-alkanes (Zhao et al. 2015, Zhao et al. 2014). This approach corrected the



signal variation of hydrocarbons in the GC-MS and was widely adopted for quantifying unresolved
complex mixtures (UCMs) (Shen et al. 2023, Zhao et al. 2022). We chose four types of standards
belonging to different compound categories with similar $1^{st}$ RTs and different $2^{nd}$ RTs, including $C_{19}H_{40}$
($1^{st}$ RT = 34.6 min, $2^{nd}$ RT = 1.03 s), 9,10-anthracenedione ($1^{st}$ RT = 36.07 min, $2^{nd}$ RT = 3.85 s), $C_{19}H_{40}$
($1^{st}$ RT = 36.54 min, $2^{nd}$ RT = 1.07 s), and fluoranthene ($1^{st}$ RT = 37.00 min, $2^{nd}$ RT = 3.04 s), and
assessed the deviation of slopes of each pair of the standards. Results in Figure 4 show that the deviation
between the three pairs of standards was 1809% ($C_{19}$ $n$-alkane vs. 9,10-anthracenedione), 1903% (9,10-
anthracenedione vs. $C_{20}$ $n$-alkane), and 105% ($C_{20}$ $n$-alkane vs. fluoranthene), respectively. The
quantitative errors in quantifying unidentified chromatographic peaks using responses of $n$-alkanes could
reach three orders of magnitude, especially for O-containing species, and those errors in quantifying
aromatic components, e.g., PAHs, also exceeded 100% in some cases.

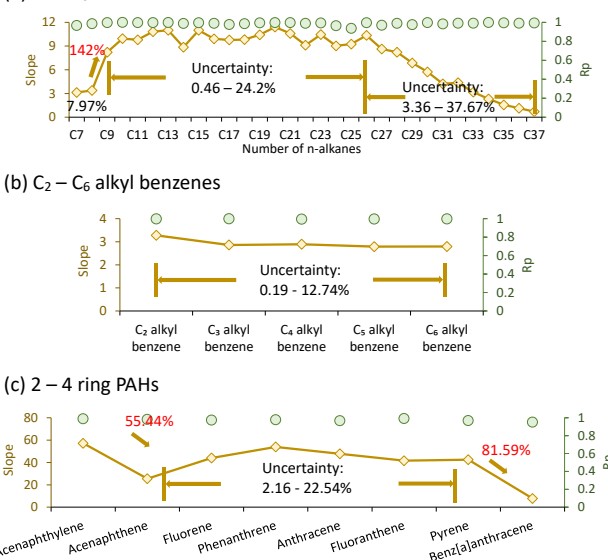


**Figure 3. Slope and Pearson correlation variation of (a) $C_7 – C_{37}$ $n$-alkanes, (b) $C_2 – C_6$ alkyl-substituted**
**benzenes, and (c) 2 – 4 ring PAHs. Brown diamond dots represent slopes of different species and are**
**referenced to the left-axis. Green circles denote the Pearson correlation of individual species and are**
**referenced to the right-axis. Pearson correlation values of $n$-alkanes, $C_2 – C_6$ alkyl-substituted benzenes, and**
**2 – 4 ring PAHs range from 0.936 to 0.999, 0.994 to 0.998, and 0.952 to 0.992, respectively. Uncertainties are**
**computed using the equation provided in the main text.**



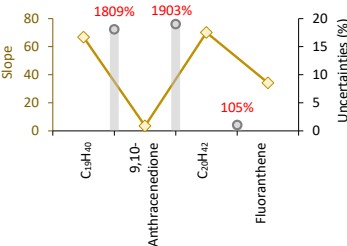


**Figure 4. Slopes and uncertainty estimation of example compounds with close 1st RTs and different 2nd RTs: $C_{19}H_{40}$ (1st RT = 34.6 min, 2nd RT = 1.03 s), 9,10-anthracenedione (1st RT = 36.07 min, 2nd RT = 3.85 s), $C_{20}H_{42}$ (1st RT = 36.54 min, 2nd RT = 1.07 s), and fluoranthene (1st RT = 37.00 min, 2nd RT = 3.04 s). Brown diamond dots represent slopes of different species and are referenced to the left-axis. Gray bars denote the uncertainty estimation of example compounds and are referenced to the right-axis.**

### 3.3 Cluster analysis in organic vapor and aerosol samples

The model was applied to organic vapor samples from HDDV tailpipe emission (HDDV vapours for short), aerosol samples from HDDV tailpipe emission (HDDV aerosols for short), and atmospheric aerosol samples (ambient aerosols for short) for cluster analysis. The results are shown in Figure S16, displaying the top few species' distribution with a contribution fraction exceeding 5%, and in Figure 5, showing the mass stacking. Overall, the speciated chromatographic peaks accounted for 85%, 82%, and 99% for HDDV vapor, HDDV aerosol, and ambient aerosol samples, respectively. The unidentified peaks were less than 20% and addressed in greater details in the supporting information (S1).

Distinct cluster distribution features could be extracted. For ambient aerosol samples, six compound clusters were filtered, and aliphatic ketones were the most abundant cluster, contributing to 27% of all the peak signals, followed by alkanes and alkenes. A notable fraction of 15.2% of organic nitros was observed in ambient samples exclusively, indicating significant secondary nitrate formation processes under atmospheric conditions. Aliphatic acids and oxy-PAHs were also detected at an abundant level, and the top six groups accounted for over 95% of the total classified peak signals. Minor but non-negligible fractions included cycloalkane, aliphatic alcohols, and phenols and alkyl-substituted phenols. Similarly, aliphatic ketones ranked first for HDDV aerosol samples, with the mass intensity reaching 46% of the total signals, followed by alkanes. Aliphatic alcohols and oxy-PAHs were detected at an abundant level, and the top four groups accounted for over 88% of the total classified peak signals. Cycloalkanes, amides, phenols and alkyl-substituted phenols, and alkenes were compound clusters with lower abundance ranging from 1-4%.



For HDDV vapours, the most abundant group was phenols and alkyl-substituted phenols, constituting

34% of the total peak signals. Compared with previous results where the most abundant group was

reported to be alkanes,(Wang et al. 2022, Alam et al. 2019) the adoption of the innovative model

contributed to resolving the oxygenated factions and reduced inaccuracies in SOA simulation due to the

lack of species information. The compound cluster is confirmed by 1) the retention time window

including 1st RT and 2nd RT,  and 2) the mass spectra. Detailed information is displayed in Figure S15.

The 2nd RTs of the identified phenols and alkyl-substituted phenols range from 1.45 to 1.78 s, well above

the hydrocarbon regions, of which the 2nd RTs fall within the range of 1.0 to 1.15 s approximately. Their

mass spectra also feature with the typical phenol ions at m/z = 94, 107, 121, 135, 149, and 191. Alkanes

ranked as the second top species, followed by $C_1$ alkyl-substituted benzene. $C_1 - C_6$ alkyl-substituted

benzenes were negligible in both ambient and HDDV aerosol samples, whereas in notable abundance in

HDDV vapor samples. This distribution aligned with their placement in the GC×GC plot, indicating they

were relatively volatile species and partitioned predominantly into the gas phase. Oxy-PAHs and

aliphatic ketones contributed to 6% of the total identified peak intensities, followed by some minor

fractions, including $C_2$ alkyl-substituted benzene, cycloalkanes, and alkenes.

The model output illustrates the overall distribution of compound clusters in various gas and aerosol

samples, providing comparative insights. Carboxylic acids indicated a higher oxidation state than other

compound clusters and were exclusively observed at a notable level in ambient samples compared with

"freshly emitted" source samples. The oxidation state of dominant compounds in HDDV samples was

comparatively low. For example, a significant ketone fraction was observed in HDDV samples, with the

majority partitioning into the aerosol phase due to the long chain carbon skeleton and thus low volatility.

Phenols and alkyl-substituted phenols were the leading species in HDDV gas samples. He (2022)

reported that the oxygenated I/SVOCs accounted for over 20% of the total I/SVOCs mass in HDDV

tailpipe emissions (He et al. 2022a).  With the refinement and improvement of model performance, e.g.,

further splitting mixed mixtures, the oxygenated fraction was elevated to over 50%.

This study highlighted the systematic presence and distribution of N-containing compounds in oxidized

valences (including nitros) and reduced valences (including amides, amines, pyridines). Among them,

amines and amides were key precursors for new particle formation processes in a polluted atmosphere

(Saeki et al. 2022, Cai et al. 2021),  and pyridines, with the N atom in the aromatic ring, were readily

dissolved in water, participating in the global N cycle in ecosystems (Kosyakov et al. 2020). Nitros





covered a wide range of organic compounds with NO or NO₂ substituents and served as critical tracers
for secondary nitrate formation processes. Amines and pyridines were volatile species occupying the
early section of the GC×GC space, while nitros and amides were distributed in the middle and rear space.
Individual N-containing species were at trace levels under atmospheric conditions and were hardly
detectable. Moreover, authentic standards or high-resolution mass spectrometry were required to identify
and quantify each compound (Zhang et al. 2018). With the establishment of an algorithmic solution, we
were able to conduct a full scan of N-containing compound clusters.
In addition to common features, specific compounds were identified in separate samples and could
potentially serve as markers or tracers for primary emission. Adamantane and its derivatives, with the
fusion of three cyclohexane rings (chemical structure and mass spectrum shown in Figure S17a), were
natural products in petroleum (Stout and Douglas 2004). They were volatile and had previously been
isolated using GC×GC-ToF-MS in crude oil (Wang et al. 2013). Adamantanes were observed in HDDV
vapor samples, contributing 1.4% to the identified peaks. Hopane (chemical structure and mass spectrum
shown in Figure S17b) was also a natural product in petroleum and bitumen, and it was an important
marker for vehicle emissions due to its persistency and stability (He et al. 2022b, Wong et al. 2021).
Hopane was reported to survive heat treatment up to 460 °C and was exclusively detected in HDDV
aerosol samples, with an intensity fraction of 0.3% (Wu and Geng 2016).

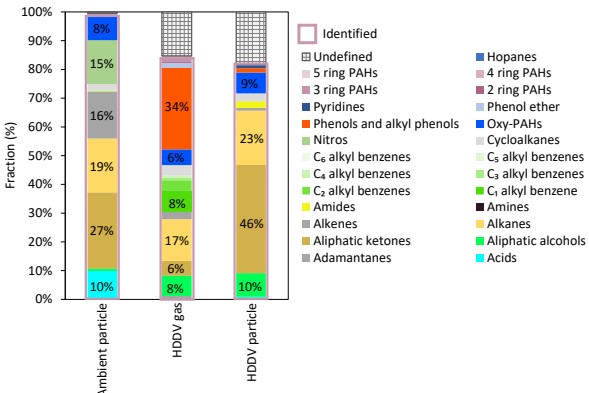


**Figure 5. The fraction distributions of different compound clusters in ambient aerosol samples, HDDV tailpipe vapours, and HDDV tailpipe aerosols. Numbers labelled on each column represent the fractions of the top few groups in different samples. Identified clusters are outlined in light purple.**



**4 Conclusions and outlook**

We presented an innovative method for optimizing the separation and identification of organic vapours and aerosols, with a focus on establishing molecular descriptors and cluster analysis algorithms. The model outputs were validated using field samples with high accuracy and integrity. Less than 20% of the peaks were unresolved components. The retention patterns of various compound groups and their distribution in the GC×GC plot were resolved, and the influence of functional groups on fragmentation was thoroughly addressed. We also provided a comprehensive analysis of the quantification uncertainties of this new approach and highlighted the significant quantitative errors when using *n*-alkanes as semi-quantification surrogates. This model was applied to various types of field samples, and the results revealed distinctive distribution patterns of compound clusters and contribution fractions, providing valuable insights into the compositions of organic vapours and aerosols, and offering potential markers for specific emission sources.

Compound speciation in atmospheric chemistry continues to be a dynamic and challenging field. Speciated compounds enable models to consider the diversity of organic species and dynamic chemical transformation in the atmosphere, contributing to more accurate SOA simulation results. It also allows for a more refined description of the dispersion of pollutants, thereby assisting in the development of localized air quality management strategies, as we strive for a more accurate and broad understanding of atmospheric chemistry.



**Supplement link:**
**Author contribution:**
X.H.: Conceptualization, formal analysis, model development, data validation, writing−original draft,
funding acquisition; X.Z.: Writing−reviewing and editing, project administration, supervision, funding
acquisition; S.G: Experiment; L.Z., T.C., B.Y., and S.X.: Experiment; Q.W., Z.L., Y.Y., S.Z., and Y.W.:
Data validation, writing−reviewing and editing.
**Competing interests:**
The authors declare that they have no conflict of interest.
**Acknowledgements**
The authors acknowledge the financial support of the National Natural Science Foundation of China
(Grant No. 42105100 and 42261160645), Scientific Research Fund at Shenzhen University (868-
000001032089 and 827-000907), and Macao Science and Technology Development Fund
(0023/2022/AFJ).

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



For Table of Contents Only

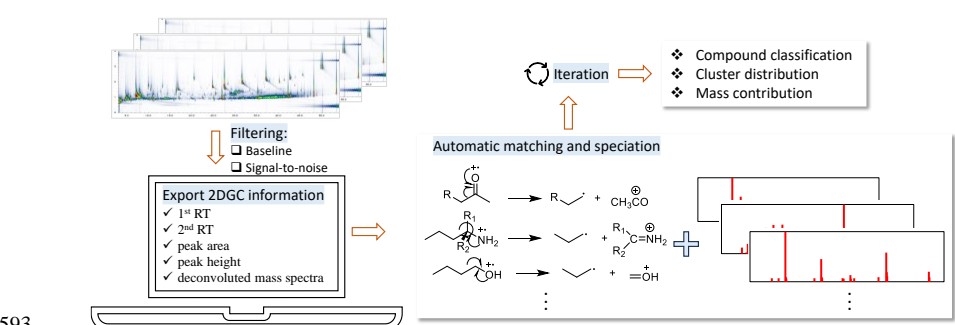
