# Peer review of "Automated compound speciation, cluster analysis, and"

_EGUsphere, 2024_

## Author Comment (AC1)

Point-to–point responses to review comments (egusphere-2024-1671)

Title: Automated compound speciation, cluster analysis, and quantification of organic vapors and aerosols using comprehensive two-dimensional gas chromatography and mass spectrometry

Review of Xiao He et al.

He et al. described an algorithm to cluster and quantify compounds measured by GC×GC. They applied this algorithm to HDDV emission and ambient $PM_{2.5}$ samples and achieved promising results. I think this work presents a novel and comprehensive approach for the analysis of complex GC×GC data. However, more details of the algorithms should be provided. Overall, I think this work can be accepted for publication after a minor revision.

Response:

We thank the reviewer for the comments and insights. The separation and detection capabilities of GC×GC allow for the speciation of complex organic compounds in gas and aerosol phases. However, the massive amount of data generated by GC×GC greatly increases threshold for its application and limits the widespread use of this technology. This work explored the possibility of combining intelligent data processing with advanced analytical techniques to enhance the detection capabilities of GC×GC and analyze complex organic compounds. We proposed a set of algorithms with the target of classifying bulk organics and performing cluster analysis. We thank the reviewer for supporting our work, and we provide below a point-by-point response to the individual comments.

Major comments

My main concern is the clustering algorithm is not described in enough detail, which makes it hard for researchers, especially those new to this field, to implement. For example, Lines 172-180 say that functional groups affect the mass spectra of compounds, and indicative reaction schemes were incorporated into the algorithm. S1 in SI described these reaction schemes. However, it is not clear how these rules are applied in the algorithm.

Response:

Thanks for the comment. As we noted in S1 in the SI, compounds containing hydrocarbon chains give rise to a series of ions separated by 14 Da ($-CH_2-$), as shown in Figure 1. As a result, the top ions to identify alkanes would be m/z = 43, m/z = 57, m/z = 71, and m/z = 84. Due to the stability of chemical groups, generally, the abundance of m/z = 57 is highest, followed by m/z = 43 and m/z = 71. When incorporating these rules into the data treatment software (Canvas, version 2.5, J&X Technologies), a few steps need to be taken, as shown in Figure 2. Four built-in features can be deployed. ABUND (X) returns the normalized abundance of the input ion mass; HASMASS (X) returns the value to indicate if the input ion exists; ORDER (X) returns the order of the input ion mass; MASS (X) returns the mass of the input ion's order. Additionally, the function allows two logical operators, "And" and "Or". Then, cluster of alkanes can be extracted by the following rules:

((MASS(1)=43 && (MASS(2)=57 || MASS(2)=71 || MASS(2)=41)) || (MASS(1)=57 && (MASS(2)=43 || MASS(2)=71 || MASS(2)=41)))

where "&&" and "||" refers to the logical operators "And" and "Or", respectively. Paste the rules in Ion Extractor Editor and the cluster of alkanes can be filtered.

For compounds with heteroatoms, the fragmentation can be complicated. As we noted in S1 in the SI, aliphatic amines often undergo cleavage at the α-C−C bond to produce relatively stable ions: $CH_2NH_2^+$ (m/z 30), $C_2H_4NH_2^+$ (m/z 44), and $C_3H_6NH_2^+$ (m/z 58) for amine groups attached to the primary, secondary, and tertiary carbons, respectively (Figure 3).[1] Then, cluster of aliphatic amines can be extracted by the following rules:

(MASS(1) = 30 && ABUND(MASS(2)) < 20) || (MASS(1) = 58 && ABUND(MASS(2)) < 40)|| (MASS(1) = 58 && MASS(2) = 59)|| (MASS(1) = 30 && (MASS(2) = 31 || MASS(2) = 28))||((ABUND(30) + ABUND(44)) > 100)

In total, 26 compound clusters were constructed with high accuracy and repeatability.

R — m/z = 43
m/z = 57
m/z = 71
m/z = 85

Figure 1. The common fragmentation patterns of n-alkanes.

Open Canvas → Open Browser → File → Load Data (load a sample) →
Speciation → Find All Peaks → Mass Spectrum → Extraction Rules

[Figure]

Figure 2. The steps to enable the ion extract function built in Canvas.

$R_1, R_2$ = H, m/z 30
$R_1$ = $CH_3$, $R_2$ = H, m/z 44
$R_1$ = $CH_3$, $R_2$ = $CH_3$, m/z 58

Figure 3. The α-C−C bond cleavage of aliphatic amines produces $CH_2NH_2^+$ (m/z 30), $C_2H_4NH_2^+$ (m/z 44), and $C_3H_6NH_2^+$ (m/z 58) for amine groups attached to the primary, secondary, and tertiary carbons.

We added the following texts and figure in the revised supporting information:

"…When incorporating these rules into the data treatment software (Canvas, version 2.5, J&X Technologies), several steps are necessary, as depicted in Figure S5. The software offers four built-in features: ABUND (X) returns the normalized abundance of the input ion mass; HASMASS (X) indicates whether the input ion exists; ORDER (X) specifies the order of the input ion mass; MASS (X) returns the mass of the input ion's order. Additionally, the function supports two logical operators, "And" and "Or". Subsequently, cluster of alkanes can be extracted by the following rules:

((MASS(1)=43 && (MASS(2)=57 || MASS(2)=71 || MASS(2)=41)) || (MASS(1)=57 && (MASS(2)=43 || MASS(2)=71 || MASS(2)=41)))

where "&&" and "||" refers to the logical operators "And" and "Or", respectively. Paste the rules in Ion Extractor Editor and the cluster of alkanes can be isolated.

Similar, the cluster of aliphatic amines can be extracted by the following rules:

(MASS(1) = 30 && ABUND(MASS(2)) < 20) || (MASS(1) = 58 && ABUND(MASS(2)) < 40)|| (MASS(1) = 58 && MASS(2) = 59)|| (MASS(1) = 30 && (MASS(2) = 31 || MASS(2) = 28))||((ABUND(30) + ABUND(44)) > 100)

In total, the extraction rules for 26 compound clusters are constructed with high accuracy and repeatability.

Open Canvas → Open Browser → File → Load Data (load a sample) → Speciation → Find All Peaks → Mass Spectrum → Extraction Rules

[Figure]

Figure S5. The steps to enable the ion extract function built in Canvas."

Lines 193-198: These descriptions should be enriched. The authors may want to provide pseudo-code like the one for the quantification algorithm in the box in Figure 1.

Response:

Thanks for the suggestion. The software offers four built-in features: ABUND (X) returns the normalized abundance of the input ion mass; HASMASS (X) indicates whether the input ion

exists; ORDER (X) specifies the order of the input ion mass; MASS (X) returns the mass of the input ion's order. Additionally, the function supports two logical operators, "And" and "Or". As we respond to the previous comment, cluster of alkanes can be extracted by the following rules:

((MASS(1)=43 && (MASS(2)=57 || MASS(2)=71 || MASS(2)=41)) || (MASS(1)=57 && (MASS(2)=43 || MASS(2)=71 || MASS(2)=41)))

where "&&" and "||" refers to the logical operators "And" and "Or", respectively. Paste the rules in Ion Extractor Editor and the cluster of alkanes can be filtered.

Cluster of aliphatic amines can be extracted by the following rules:

(MASS(1) = 30 && ABUND(MASS(2)) < 20) || (MASS(1) = 58 && ABUND(MASS(2)) < 40)|| (MASS(1) = 58 && MASS(2) = 59)|| (MASS(1) = 30 && (MASS(2) = 31 || MASS(2) = 28))||((ABUND(30) + ABUND(44)) > 100)

We add the following descriptive pseudo-codes in the revised supporting information:

"…The scripts began by recognition of the common mass spectra features of compound cluster of interest and are addressed in more details in the following descriptive framework:

…

```
For (i = 1 to m) # m equals the number of all tested samples.
  Load the sample
  Peak identification
  Baseline correction
  Mass spectra deconvolution
  Peak smoothing
   For (j = 1 to 26) # In total, 26 compound clusters were constructed with high accuracy and
repeatability.
     Execute the Extraction rule of cluster (j)
     Export peak number, 1st RT and 2nd RT, peak area, peak height, peak width, and deconvoluted
mass spectra
    Next j
  Next i
```
"

Minor comments

Line 56: Should write NIST instead of NIST20 because NIST20 only refers to the 2020 version.

Response:

Suggestion taken.

Line 57: Retention Index matching was introduced decades ago, not by Zang et al.

Response:

Thanks for the comments. We then removed the reference in the revised text to avoid ambiguity.

Line 88: What does "retention rate" mean here?

Response:

Retention rate is a statistical measurement of the number of people or products that remain involved in some kind of entity. In this context, the retention rate of HDDV refers to the proportion of HDDVs in possession among the total number of vehicles.

Lines 113-115: The authors may want to write down how many samples were collected (or used in this work).

Response:

In total, we collected 55 TA tube samples (11 of them were field blank samples), 20 HDDV aerosol samples (3 of them were field blank samples), and 6 ambient aerosol samples (one blank sample). We add the following texts in the revised manuscript:

"…Sampling strategy followed a regular schedule of one 24-h sample every day using a high-volume sampler (Th-1000c II, Wuhan Tianhong Environmental Protection Industry Co., Ltd). In total, 55 TA tube samples (including 11 field blank samples), 20 HDDV aerosol samples (including 3 field blank samples), and 6 ambient aerosol samples (including one blank sample) were collected. TD samples were kept dry at room temperature, and quartz filters were stored frozen at −18 °C before analysis. All sampling materials were pre-baked thoroughly to remove potential carbonaceous contamination."

Line 122: What do you mean by TD samples? Do you mean sorbent tubes? I would suggest renaming it because filter samples also went through thermal desorption. Also, why were the TD samples kept at room temperature? Won't the analytes evaporate?

Response:

Thanks for the suggestion. We now rephrase the statement in the revised manuscript:

"…In total, 55 TA tube samples (including 11 field blank samples), 20 HDDV aerosol samples (including 3 field blank samples), and 6 ambient aerosol samples (including one blank sample) were collected. The sorbent tubes were well sealed and kept dry at room temperature, and quartz filters were stored frozen at −18 °C before analysis. All sampling materials were pre-baked thoroughly to remove potential carbonaceous contamination."

According to the user manual, the thermal desorption tube is sealed well, and samples can be preserved for at least two years in it at room temperature. We then complement this information in the revised manuscript.

Line 145: Should give more description of the mass spectrometer, like resolution and ion source.

Response:

Thanks for the suggestion. We provide more description of the mass spectrometer in the revised manuscript:

"The MS had an integer resolution and was conducted in electron impact positive (EI+) mode (70 eV). It was operated over a range of 20–350 amu, and the temperature of the transfer line, ion source, and MS quadrupole was 300 °C, 250 °C, and 170 °C, respectively."

Line 155: The authors may want to list the deuterated internal standards in the SI.

Response:

Suggestion taken. The deuterated internal standards were listed in Table S1 in the revised supporting material.

Table S1. List of the deuterated internal standards.

| Internal Standard | Formula |
|---|---|
| $n$-Dodecane-d$_{26}$ | $C_{12}D_{26}$ |
| $n$-Hexadecane-d$_{34}$ | $C_{16}D_{34}$ |
| $n$-Eicosane-d$_{42}$ | $C_{20}D_{42}$ |
| $n$-Tetracosane-d$_{50}$ | $C_{24}D_{50}$ |
| $n$-Octacosane-d$_{58}$ | $C_{28}D_{58}$ |
| Naphthalene-d$_8$ | $C_{10}D_8$ |
| Acenaphthene-d$_{10}$ | $C_{12}D_{10}$ |
| Phenanthrene-d$_{10}$ | $C_{14}D_{10}$ |
| Chrysene-d$_{12}$ | $C_{18}D_{12}$ |

We add the following texts in the revised manuscript:

"…TD samples were injected with 2 µL of deuterated internal standard (IS) mixing solution through a mild $N_2$ blow (CSLR, Markes International). The list of deuterated IS is shown in Table S1."

Line 159: Would Retention Index be a better chromatographic variable to use in cluster matching?

Response:

The quick answer is yes. As we already know, Retention Index (RI) of a compound is calculated by comparing its retention time to the retention times of a series of standard compounds (typically n-alkanes). RI is a useful metric in chromatographic analysis. It helps to identify compounds based on their elution characteristics. When constructing this method, we adopted RI to refine the cluster analysis results to eliminate the ambiguity introduced by different runs and tests, as shown in Figure 1. However, we make it clear that, according to our experimental records, the retention times of the target analytes did not change much under fixed conditions.

Lines 227-229: The authors cited Franklin et al. to suggest that the decomposition products may not be very significant. But in that work, Emily derivatized the polar compounds with MSTFA, which helps to protect the thermally labile compounds. I think the comparison with her work here could be misleading. It is better for the authors to delete this comparison.

Response:

Thanks for the suggestion. We remove the comparison with the referred work in the revised manuscript.

Line 233: How is this correlation calculated?

Response:

Correlation relationships were built by spiking gradient volumes (0, 1, 2, 5, 7, 10 μL) of working solution plus 2 μL of internal standard solution and establishing a linear relationship between the peak area (PA) ratio (PA of each authentic standard/PA of the corresponding internal standard) and the spiked mass. The description of calibration curves can be found in lines 207-212.

We also add the following texts in the revised manuscript:

"Strong linear correlations were achieved for this set of authentic standards between the peak area ratio and the spiked mass, with Pearson's R ranging from 0.97 to 0.99."

Section 3.2 Uncertainty Estimation: I feel this section is only about the uncertainty associated with (semi-)quantification of compounds correctly clustered, not about the uncertainty associated with clustering. The authors may want to clarify.

Response:

Thanks for the suggestion. We corrected the subtitle of section 3.2 in the revised manuscript:

"3.2 Estimation of the uncertainty associated with the (semi-) quantification"

Line 356: S2?

Response:

Sorry for the mistake. It should be S2 not S1.

Line 358: What do you mean by "filtered"?

Response:

Sorry for the typo errors. I delete the words in the revised manuscript.

Line 358: Cooking can also contribute substantially to carboxylic acids in ambient aerosol in urban areas like Shenzhen.

Response:

This is exactly the case. Cooking is a significant source for carboxylic acids in ambient aerosol in urban cities, especially fatty acids such as oleic acid, palmitic acid, and stearic acid. Extensive studies in field measurements and source apportionment have reported this. For example, Wang et al. (2020) demonstrated the use of high-time-resolution fatty acid markers to capture the dynamic changes of cooking emissions.[2]

References:

(1) Mikaia, A. Protocol for Structure Determination of Unknowns by EI Mass Spectrometry. I. Diagnostic Ions for Acyclic Compounds with up to One Functional Group. *Journal of Physical and Chemical Reference Data* **2022**, *51* (3), 031501. DOI: 10.1063/5.0091956.
(2) Wang, Q.; He, X.; Zhou, M.; Huang, D. D.; Qiao, L.; Zhu, S.; Ma, Y.-G.; Wang, H.-L.; Li, L.; Huang, C.; et al. Hourly Measurements of Organic Molecular Markers in Urban Shanghai, China: Primary Organic Aerosol Source Identification and Observation of Cooking Aerosol Aging. *ACS Earth and Space Chemistry* **2020**, *4* (9), 1670-1685. DOI: 10.1021/acsearthspacechem.0c00205.

---

## Author Comment (AC2)

Point-to–point responses to review comments (egusphere-2024-1671)

Title: Automated compound speciation, cluster analysis, and quantification of organic vapors and aerosols using comprehensive two-dimensional gas chromatography and mass spectrometry

Review of Xiao He et al.

The development of analytical techniques has put forward higher requirements for the identification and processing of complex organic species. Developing innovative data parsing methods important to understand intricate chemical mixtures. The study reported an innovative method for the semi-automated identification and quantification of complex organic mixtures using GC×GC-MS and applied this method to organic vapor and aerosol samples collected from tailpipe emissions of heavy-duty diesel vehicles and the ambient atmosphere. The study is novelty, providing an automated approach for chemical compound speciation and cluster analysis. The manuscript is well organized and written. I recommend it can be accepted after a minor revision

Thanks very much for the positive evaluation of this work. Our point-to–point responses to your comments are presented below. Our response texts are marked in blue in this document. The revised texts in the main manuscript and the supporting information are also marked in blue.

**Specific comments:**

Line 88: "Despite a low retention rate". What is retention rate? Do the authors mean population rate in the whole vehicle fleet?

Response:

Yes. Retention rate is a statistical measurement of the number of people or products that remain involved in some kind of entity. In this context, the retention rate of HDDV refers to the proportion of HDDVs in possession among the total number of vehicles.

Line 105: Section "2.1 Sample collection, treatment, and instrumental analysis". Detailed sample information was not available in this section. For example, sampling season and the relevant PM2.5 concentration in the atmosphere were unclear. How many diesel vehicles were measured, and their emission levels, engine size, repetition frequency, etc.?

Response:

Thanks for the suggestion. For the collection of HDDV tailpipe emissions, two HDDVs equipped with the selective catalytic reduction (SCR) system were recruited. The two HDDVs met the China IV national emission standard and were manufactured in 2021. More information is summarized in Table 1. In total, we collected 55 TA tube samples (11 of which were field blank samples) and 20 HDDV aerosol samples (3 of which were field blank samples). 6 ambient aerosol samples (including one blank sample) were collected during November 2023 and the relevant PM concentrations are listed in Table 2.

Table 1. Information of the two test HDDVs.

| Vehicle ID | Emission standard | Aftertreatment | Model year | Gross vehicle weight (kg) | Vehicle type | Mileage (× 10³ km) | Engine model |
|---|---|---|---|---|---|---|---|
| #1 | China V | Selective catalytic reduction | 2021 | 25000 | Semi-trailer tractor | 22.2 | dCi450-51 |
| #2 | China V | Selective catalytic reduction | 2021 | 25000 | Semi-trailer tractor | 34.8 | MC13.54-50 |

Table 2. List of ambient samples and the corresponding PM concentration.

| Sample ID | Collection date | PM$_{2.5}$ concentration (µg/m$^3$) | PM$_{10}$ concentration (µg/m$^3$) |
|---|---|---|---|
| #1 | 1-Nov-2023 | 17 | 38 |
| #2 | 5-Nov-2023 | 14 | 28 |
| #3 | 10-Nov-2023 | 16 | 35 |
| #4 | 11-Nov-2023 | 13 | 28 |
| #5 | 17-Nov-2023 | 19 | 66 |
| Blank | 17-Nov-2023 | 19 | 66 |

We add the following texts in the revised manuscript:

"…following the China heavy-duty commercial vehicle test cycle for tractor trailers (CHTC-TT) driving cycles. Two HDDVs equipped with the selective catalytic reduction (SCR) system were recruited. The two HDDVs met the China IV national emission standard and were manufactured in 2021. More information is summarized in Table S1. The average temperature…"

"…building on the campus of Shenzhen University (22.60°N, 114.00°E) during November 2023 in western Shenzhen…"

"…every day using a high-volume sampler (Th-1000c II, Wuhan Tianhong Environmental Protection Industry Co., Ltd). In total, 55 TA tube samples (including 11 field blank samples), 20 HDDV aerosol samples (including 3 field blank samples), and 6 ambient aerosol samples (including one blank sample) were collected. The list of ambient samples and the relevant PM concentrations are listed in Table S2. The sorbent tubes were well sealed and stored…"

We add the following texts in the revised supporting information:

"Table S1. Information of the two test HDDVs.

| Vehicle ID | Emission standard | Aftertreatment | Model year | Gross vehicle weight (kg) | Vehicle type | Mileage (× 10³ km) | Engine model |
|---|---|---|---|---|---|---|---|
| #1 | China V | Selective catalytic reduction | 2021 | 25000 | Semi-trailer tractor | 22.2 | dCi450-51 |

| | | | | | | | |
|---|---|---|---|---|---|---|---|
| #2 | China V | Selective catalytic reduction | 2021 | 25000 | Semi-trailer tractor | 34.8 | MC13.54-50 |

Table S2. List of ambient samples and the corresponding PM concentration.

| Sample ID | Collection date | $PM_{2.5}$ concentration ($\mu g/m^3$) | $PM_{10}$ concentration ($\mu g/m^3$) |
|---|---|---|---|
| #1 | 1-Nov-2023 | 17 | 38 |
| #2 | 5-Nov-2023 | 14 | 28 |
| #3 | 10-Nov-2023 | 16 | 35 |
| #4 | 11-Nov-2023 | 13 | 28 |
| #5 | 17-Nov-2023 | 19 | 66 |
| Blank | 17-Nov-2023 | 19 | 66 |

"

Line 109-111: "The average temperature in the sampling train was precisely controlled at 47 °C, and airflow, relative humidity, and airflow, relative humidity, and pressure were monitored simultaneously". "and airflow, relative humidity" were repeated.

Response:

Sorry for the typo errors. We revise them in the revised manuscript.

Line 163: Section "2.3 Algorithmic development". I think the methodology how the authors train, iterate, and optimize the scripts was introduced somewhat roughly, which is important whether this algorithm can be referenced by other studies. For example, how many parameters does the algorithm contain and how many parameters can be optimized, what about their impacts. How many times did the authors conduct training, how effective was the training, and so on.

Response:

Thanks for this insightful comment. All data utilized to develop and test the scripts were processed by Canvas Browser (version 2.5, J&X Technologies) for basic preprocessing, such as baseline correction, mass spectra deconvolution, and peak smoothing. The software offers four built-in features: ABUND (X) returns the normalized abundance of the input ion mass; HASMASS (X) indicates whether the input ion exists; ORDER (X) specifies the order of the input ion mass; MASS (X) returns the mass of the input ion's order. It also supports two logical operators, "And" and "Or".

Basically, compounds containing hydrocarbon chains give rise to a series of ions separated by 14 Da (-$CH_2$-), as shown in Figure 1. As a result, the top ions to identify alkanes would be m/z = 43, m/z = 57, m/z = 71, and m/z = 84. Due to the stability of chemical groups, generally, the abundance of m/z = 57 is highest, followed by m/z = 43 and m/z = 71. When incorporating these rules into the data treatment software, a few steps need to be taken, as shown in Figure 2. Cluster of alkanes can be extracted by the following rules:

((MASS(1)=43 && (MASS(2)=57 || MASS(2)=71 || MASS(2)=41)) || (MASS(1)=57 && (MASS(2)=43 || MASS(2)=71 || MASS(2)=41)))

where "&&" and "||" refers to the logical operators "And" and "Or", respectively. Paste the rules in Ion Extractor Editor and the cluster of alkanes can be filtered.

For compounds with heteroatoms, the fragmentation can be complicated. Taking aliphatic amines as an example, they often undergo cleavage at the α-C−C bond to produce relatively stable ions: $CH_2NH_2^+$ (m/z 30), $C_2H_4NH_2^+$ (m/z 44), and $C_3H_6NH_2^+$ (m/z 58) for amine groups attached to the primary, secondary, and tertiary carbons, respectively (Figure 3).[1] Then, cluster of aliphatic amines can be extracted by the following rules:

(MASS(1) = 30 && ABUND(MASS(2)) < 20) || (MASS(1) = 58 && ABUND(MASS(2)) < 40)|| (MASS(1) = 58 && MASS(2) = 59)|| (MASS(1) = 30 && (MASS(2) = 31 || MASS(2) = 28))||((ABUND(30) + ABUND(44)) > 100)

Similarly, a total of 26 compound clusters were constructed with high accuracy and repeatability.

R⌐⌐⌐⌐⌐⌐⌐⌐  m/z = 43
          m/z = 57
          m/z = 71
          m/z = 85

Figure 1. The common fragmentation patterns of n-alkanes.

Open Canvas → Open Browser → File → Load Data (load a sample) →
Speciation → Find All Peaks → Mass Spectrum → Extraction Rules

[Figure]

Figure 2. The steps to enable the ion extract function built in Canvas.

$R_1, R_2$ = H, m/z 30
$R_1$ = $CH_3$, $R_2$ = H, m/z 44
$R_1$ = $CH_3$, $R_2$ = $CH_3$, m/z 58

Figure 3. The α-C−C bond cleavage of aliphatic amines produces $CH_2NH_2^+$ (m/z 30), $C_2H_4NH_2^+$ (m/z 44), and $C_3H_6NH_2^+$ (m/z 58) for amine groups attached to the primary, secondary, and tertiary carbons.

We added the following texts and figure in the revised supporting information:

"…When incorporating these rules into the data treatment software (Canvas, version 2.5, J&X Technologies), several steps are necessary, as depicted in Figure S5. The software offers four built-in features: ABUND (X) returns the normalized abundance of the input ion mass; HASMASS (X) indicates whether the input ion exists; ORDER (X) specifies the order of the input ion mass; MASS (X) returns the mass of the input ion's order. Additionally, the function supports two logical operators, "And" and "Or". Subsequently, cluster of alkanes can be extracted by the following rules:

$((MASS(1)=43$ && $(MASS(2)=57$ || $MASS(2)=71$ || $MASS(2)=41))$ || $(MASS(1)=57$ && $(MASS(2)=43$ || $MASS(2)=71$ || $MASS(2)=41)))$

where "&&" and "||" refers to the logical operators "And" and "Or", respectively. Paste the rules in Ion Extractor Editor and the cluster of alkanes can be isolated.

Similar, the cluster of aliphatic amines can be extracted by the following rules:

$(MASS(1) = 30$ && $ABUND(MASS(2)) < 20)$ || $(MASS(1) = 58$ && $ABUND(MASS(2)) < 40)$|| $(MASS(1) = 58$ && $MASS(2) = 59)$|| $(MASS(1) = 30$ && $(MASS(2) = 31$ || $MASS(2) = 28))$||$((ABUND(30) + ABUND(44)) > 100)$

In total, the extraction rules for 26 compound clusters are constructed with high accuracy and repeatability.

Open Canvas → Open Browser → File → Load Data (load a sample) → Speciation → Find All Peaks → Mass Spectrum → Extraction Rules

[Figure]

Figure S5. The steps to enable the ion extract function built in Canvas."

To better describe the algorithm, we also add the following descriptive pseudo-codes in the revised manuscript:

"…The scripts began by recognition of the common mass spectra features of compound cluster of interest and are addressed in more details in the following descriptive framework:

…

For (i = 1 to m) # m equals the number of all tested samples.
    Load the sample
    Peak identification

```
    Baseline correction
    Mass spectra deconvolution
    Peak smoothing
      For (j = 1 to 26) # In total, 26 compound clusters were constructed with high accuracy
   and repeatability.
        Execute the Extraction rule of cluster (j)
        Export peak number, 1st RT and 2nd RT, peak area, peak height, peak width, and
   deconvoluted
   mass spectra
      Next j
   Next i
```
"

Line 286: Section "3.2 Model uncertainty estimation". The authors have conducted a detail uncertainty analysis on the model estimation. However, I still wonder the differences of the results analyzed by this new approach compared to the traditional one. It would be better if there could be some validation for some species by two different identification methods.

Response:

Thanks for the comments. Taking Octanal (Formula: $C_8C_{16}O$) as an example, it elutes at ($1^{st}$ RT = 11.1908 min, $2^{nd}$ RT = 1.295 s), and the nearest n-alkane reference compound is $C_{10}H_{22}$ ($1^{st}$ RT = 11.1923 min, $2^{nd}$ RT = 0.988 s). The chromatographic information of the two compounds is listed in Table 3. Dividing the whole chromatogram into bins based on the $1^{st}$ RT, Octanal would be assigned to B10, where $C_{10}$ n-alkane is the reference compound.[2, 3] In this case, the concentration of Octanal would be overestimated by over 210%.

Table 3. The chromatographic information of Octanal and $C_{10}H_{22}$.

| | $1^{st}$ RT (min) | $2^{nd}$ RT (s) | Slope | Group |
|---|---|---|---|---|
| $C_{10}H_{22}$ | 11.1923 | 0.988 | 9.93 | Alkane |
| $C_8H_{16}O$ | 11.1908 | 1.295 | 21.5 | Aliphatic ketone |

Line 417: Figure 5. How many samples for the heavy-duty diesel vehicle emissions and the ambient atmosphere and what about the consistency between the samples of the diesel vehicle and ambient samples, respectively?

Response:

In total, we collected 55 TA tube samples (11 of them were field blank samples), 20 HDDV aerosol samples (3 of them were field blank samples), and 6 ambient aerosol samples (one blank sample). We add the following texts in the revised manuscript:

"…Sampling strategy followed a regular schedule of one 24-h sample every day using a high-volume sampler (Th-1000c II, Wuhan Tianhong Environmental Protection Industry Co., Ltd). In total, 55 TA tube samples (including 11 field blank samples), 20 HDDV aerosol samples (including 3 field blank samples), and 6 ambient aerosol samples (including one blank sample)

were collected. TD samples were kept dry at room temperature, and quartz filters were stored frozen at −18 °C before analysis. All sampling materials were pre-baked thoroughly to remove potential carbonaceous contamination."

Since the HDDV samples and ambient samples were collected under different driving and atmospheric conditions, the sample-by-sample consistency was not compared. However, we traced the peak areas of representative deuterated internal standards, and the results were displayed in Figure S7 (Figure S8 in the revised supporting information document). The coefficients of variation (CVs) indicated the general variation of the ISs within the entire measurement.[4] Excellent stability was clearly observed, with all CV values being lower than 20%, which demonstrated the robustness of the testing system.

Reference:
(1) Mikaia, A. Protocol for Structure Determination of Unknowns by EI Mass Spectrometry. I. Diagnostic Ions for Acyclic Compounds with up to One Functional Group. *Journal of Physical and Chemical Reference Data* **2022**, *51* (3), 031501. DOI: 10.1063/5.0091956.
(2) Zhao, Y.; Hennigan, C. J.; May, A. A.; Tkacik, D. S.; de Gouw, J. A.; Gilman, J. B.; Kuster, W. C.; Borbon, A.; Robinson, A. L. Intermediate-volatility organic compounds: a large source of secondary organic aerosol. *Environ Sci Technol* **2014**, *48* (23), 13743-137550. DOI: 10.1021/es5035188.
(3) Zhao, Y.; Nguyen, N. T.; Presto, A. A.; Hennigan, C. J.; May, A. A.; Robinson, A. L. Intermediate Volatility Organic Compound Emissions from On-Road Diesel Vehicles: Chemical Composition, Emission Factors, and Estimated Secondary Organic Aerosol Production. *Environ Sci Technol* **2015**, *49* (19), 11516-11526. DOI: 10.1021/acs.est.5b02841.
(4) He, X.; Wang, Q.; Huang, X. H. H.; Huang, D. D.; Zhou, M.; Qiao, L.; Zhu, S.; Ma, Y.-g.; Wang, H.-l.; Li, L.; et al. Hourly measurements of organic molecular markers in urban Shanghai, China: Observation of enhanced formation of secondary organic aerosol during particulate matter episodic periods. *Atmospheric Environment* **2020**, *240*, 117807. DOI: 10.1016/j.atmosenv.2020.117807.